# Nano-Restoration for Sustaining Soil Fertility: A Pictorial and Diagrammatic Review Article

**DOI:** 10.3390/plants11182392

**Published:** 2022-09-14

**Authors:** Hassan El-Ramady, Eric C. Brevik, Zakaria F. Fawzy, Tamer Elsakhawy, Alaa El-Dein Omara, Megahed Amer, Salah E.-D. Faizy, Mohamed Abowaly, Ahmed El-Henawy, Attila Kiss, Gréta Törős, József Prokisch, Wanting Ling

**Affiliations:** 1Soil and Water Department, Faculty of Agriculture, Kafrelsheikh University, Kafr El-Sheikh 33516, Egypt; 2Institute of Animal Science, Biotechnology and Nature Conservation, Faculty of Agricultural and Food Sciences and Environmental Management, University of Debrecen, 138 Böszörményi Street, 4032 Debrecen, Hungary; 3College of Agricultural, Life, and Physical Sciences, Southern Illinois University, Carbondale, IL 62901, USA; 4Vegetable Crops Department, Agricultural and Biological Research Institute, National Research Centre, 33 El Buhouth St., Dokki, Giza 12622, Egypt; 5Agriculture Microbiology Department, Soil, Water and Environment Research Institute (SWERI), Sakha Agricultural Research Station, Agriculture Research Center, Kafr El-Sheikh 33717, Egypt; 6Soil Improvement and Conservation Department, Water and Environment Research Institute, Agricultural Research Center, Giza 12622, Egypt; 7Knowledge Utilization Center of Agri-Food Industry, University of Debrecen, Böszörményi út 138, 4032 Debrecen, Hungary; 8Institute of Organic Contaminant Control and Soil Remediation, College of Resources and Environmental Sciences, Nanjing Agricultural University, Nanjing 210095, China

**Keywords:** soil–plant nexus, soil degradation, soil conservation, waterlogged soil, salt-affected soil, polluted soil, degraded soil

## Abstract

Soil is a real treasure that humans cannot live without. Therefore, it is very important to sustain and conserve soils to guarantee food, fiber, fuel, and other human necessities. Healthy or high-quality soils that include adequate fertility, diverse ecosystems, and good physical properties are important to allow soil to produce healthy food in support of human health. When a soil suffers from degradation, the soil’s productivity decreases. Soil restoration refers to the reversal of degradational processes. This study is a pictorial review on the nano-restoration of soil to return its fertility. Restoring soil fertility for zero hunger and restoration of degraded soils are also discussed. Sustainable production of nanoparticles using plants and microbes is part of the process of soil nano-restoration. The nexus of nanoparticle–plant–microbe (NPM) is a crucial issue for soil fertility. This nexus itself has several internal interactions or relationships, which control the bioavailability of nutrients, agrochemicals, or pollutants for cultivated plants. The NPM nexus is also controlled by many factors that are related to soil fertility and its restoration. This is the first photographic review on nano-restoration to return and sustain soil fertility. However, several additional open questions need to be answered and will be discussed in this work.

## 1. Introduction

The soil system represents one of the main natural resources that supplies human needs for food, feed, fiber, fuel, and more [1,2]. Agroecosystems are crucial to guaranteeing human life because of the interactions among their compartments, which include soil, water, plants, microbes, humans, and animals [3]. Thus, many studies have focused on the role of agroecosystems in sustaining and restoring soil fertility, or the ability of the soil to provide needed nutrients to crops. The loss of soil fertility may result from degradational processes such as pollution of the soil–plant–water system [4], alkalinity and salinity [5], or antagonisms from other nutrients or elements that may be added during agricultural management [6]. One of the most active portions of an agroecosystem is the soil microbes. These microbes have crucial impacts, mainly in the rhizosphere, through significant reactions in soil–plant–microbial activity, which may enhance soil fertility [7]. Several reactions occur in the rhizosphere that involve the release of root exudates or plant metabolites to support the soil microbial community for transformation of nutrients [7]. Therefore, there is an urgent need to build soil organic content and microbial communities to achieve both soil fertility and sustainable agriculture [3,7,8,9,10].

A number of studies have highlighted factors that are associated with interactions among different compartments of the agroecosystem. The relationship among different nexuses and their link to soil and water has been widely investigated, such the systems of soil–water–climate change [11], water–land–energy–food [12,13], soil–food–environment–health [14], sustainable water–energy–environment [15], water–food–energy–climate [16], water–energy–waste [17], water–energy–food [18,19,20], water–food–land–ecosystem [21], water–energy–carbon [22], soil health–human health [23], and soil–water–plant–human [24]. Photographic or pictorial articles can be highly effective at communicating these complex nexuses [2]. This has led to several recent articles that have used illustrations and/or photographs to highlight topics such as smart agriculture [25], soil and humans [26], management of salt-affected soils [27], the comparison between higher plants and mushrooms [28], nano-farming [29], nano-grafting [30], and the soil–water–plant–human nexus [24]. The main difference between a typical review article and pictorial review is that the pictorial review heavily depends on presenting the available information using photographs, diagrams, and other image-based methods. The fundamental idea of this kind of presentation is that one photo or other image may be better than 1000 words, and therefore a well-illustrated review facilitates communicating the main ideas in the paper.

Therefore, this is the first photographic and diagrammatic review on nano-restoration as a means to sustain soil fertility. This work also discusses one of the most important nexuses, the nanoparticle–plant–microbe (NPM) system and its potential to restore soil fertility, focusing on the NPM nexus and its importance for sustainable agriculture. 

## 2. Methodology of the Review

The main sources for this review are articles from the major publishers (e.g., PubMed, Frontiers, ScienceDirect, Springer, MDPI, Google Scholar). The following keywords or phrases were searched: “Restoring soil fertility and zero hunger”, “main soil restoration forms”, “approaches of soil restoration”, “Sustainable synthesis of nanoparticles by plants and microbes”, “Nano-enhanced materials for soil fertility restoration”, and “Nanoparticle–plant–microbe nexus and soil fertility”. Different combinations of these keywords and phrases were entered into search engines such as “Nanoparticle and Plant”, “Nanoparticle and Microbe”, “Nanoparticle and Soil Fertility”, “Soil and Nanoparticle”, “Plant and Microbe”, “Microbe and Soil Fertility”, “Soil–Plant–Water nexus”, “Soil–Plant–Microbe Nexus”, “Soil–Water–Nanoparticle Nexus”, “Soil–Plant–Water–Microbe Nexus” and “Soil–Plant–Microbe–Nanoparticle Nexus”. The selection of articles from different engines should depend on certain criteria, mainly the reputation and quality of journals and limited publication period. Manuscripts published in the last 5 years (2018–2022) were prioritized within this review. 

## 3. Soil and the Sustainable Development Goals (SDGs)

The need to feed around 10 billion people by 2050 represents a great challenge for the entire world. This necessitates an increase in agricultural production of ~70% by 2050 [31]. Soil is a major factor in this production. Soil is central to many of the Sustainable Development Goals (SDGs) [32], as seen in Figure 1. The SDGs were announced by the United Nations and have direct and indirect impacts on managing soil functions [33]. Many of the SDGs can be directly (SDGs 2, 3, 6, 13, 15, and 17) or indirectly (SDGs 1, 5, 6, 8, 10, and 16) affected by soil quality and management [34].

Thus, the quality and persistence of soil functions and their achievement for these goals mainly rely on soil health [35]. In line with these goals, there is an urgent need for continuous support of soil ecosystem services [33]. Any progress in achieving the SDGs requires sustainable management of soils, because many SDGs are directly influenced by the properties and processes of soils [32]. 

Soil restoration is a crucial approach to achieve the goal of zero hunger [36,37] (Figure 2). Soil restoration is a process in which the reduced soil fertility or soil health/quality of degraded soil is reversed through management practices to restore ecosystem functions and services. The main things that need to be restored include (1) physical degradation (e.g., compaction, erosion, sealing, loss of structure), (2) chemical degradation (e.g., salt-affected soils, pollution, acidification), (3) biological degradation (loss of soil biodiversity, low soil organic matter), and (4) ecological degradation (loss of nutrients and carbon, inhibited in the denaturing of pollutants) [31]. There is a strong relationship between soil fertility and its management (from one side) and SDG 2 (zero hunger), from the other. Zero hunger is strongly connected to global issues represented in food security, malnutrition, and sustainable agriculture. These issues mainly depend on soil fertility and its management through the ecological management of nutrients, which is needed to overcome environmental obstacles such as soil degradation, water pollution, and climate change [38]. 

## 4. Restoration of Degraded Soils

Soil degradation can be defined as “a change in the soil health status resulting in a diminished capacity of the ecosystem to provide goods and services for its beneficiaries” [40]. Soil degradation includes losses in soil biodiversity, productivity, and fertility. The main causes of soil degradation are pollution resulting from industrial, agricultural, and commercial activities, loss of arable lands due to overgrazing, urban sprawl/expansion, climatic changes, and unsustainable agricultural practices. Restoration of soil fertility can be achieved through sustainable management of degraded lands, such as climate change mitigation through the cultivation of bioenergy crops, production of animal proteins through intensive rotational grazing, and restoration of biodiversity by converting degraded croplands into conservation plantings [38]. Soil fertility can be restored by applying different approaches as presented in Figure 3, which may include using plant growth promoting rhizobacteria and arbuscular mycorrhizal fungi, applying organic amendments, inorganic fertilization, nanomaterials and nano-nutrients, cover crops and soil surface mulching, preventing hardening or compaction of the soil, integrated application of fertilizers to include organic, inorganic and biofertilizer, perennialization of cropping systems, and enhancing the sources of ecosystem services [38,39]. Several kinds of degraded soils are well-known, such as sandy soils in arid regions, waterlogged soils, polluted soils, mined soils, and salt-affected soils. In the following sub-sections, a certain concern will focus on two common types of degraded soils (i.e., saline sandy and saline–sodic soils).

### 4.1. Saline Sandy Soils

Restoration of saline sandy soils especially in arid regions depends on the main problem of these soils (i.e., salinity level in soil, low content of organic matter and nutrients, low ability to hold water). Applying the organic amendments like compost or organic fertilizers or green manure are the most common practices in sandy soils (Figure 4). Under water stress conditions, foliar application using salicylic acid (150 mg L^−1^), and ascorbic acid (100 mg L^−1^) can support the productivity of olive trees grown in Matrouh, Egypt [41]. The integrated inoculation of pearl millet by mycorrhizae fungi, with combined application of humic acid (38.4 kg ha^−1^) and phosphoric acid (1.5 mL L^−1^) improved the availability of nutrient status of sandy calcareous soil in Mariout, southwest of Alexandria, Egypt [42]. Integrated management of K-additives (apply *Amphora* extract of algae, biochar, and compost) to improve Zucchini productivity grown on sandy soil [43]. The combined amending sandy soils with mixed organic and mineral as N-sources and irradiating seeds of faba bean to increase the crop productivity was reported by Farid et al. [44]. The microbial mixtures (*Bacillus subtilis*, *Pseudomonas flourescens*, *Pleurotus ostreatus*, and mycorrhizeen^®^) modified soil physio-chemical properties and its fertility, and consequently increased productivity of *Hibiscus sabdariffa* L. in sandy soil [45].

Soil microbes (biofertilizers) can increase soil fertility through enhancing solubility and uptake of nutrients in soil by cultivated plants, and then increase productivity and its yield [46]. The tripartite interaction among soil–plant–microbes is very important for soil fertility and sustainable agriculture. The reason represents in the nature of this relationship between plant and microbes, which lead to converting the unavailable nutrients in soil into available and uptakeable by plants [47]. Besides the acquisition of nutrients and due to the beneficial activities of soil–nutrient–microbe–plant interactions, soil microbes can also inhibit plant pathogens and induce plant defense response [47]. Under the circular economy, using agri-based organic wastes in producing bio-organic fertilizer and compost via soil beneficial microbes at the farm level are a crucial approach for a sustainable design of new cropping systems, and for increasing soil natural suppressiveness to soil-borne plant pathogens [48]. 

### 4.2. Saline–Sodic Soils

Salt-affected soils are a common problem. Salts are a major constraint for high crop productivity on about 1125 million hectares globally and are especially problematic in arid and semi-arid regions [49]. These soils are formed by both anthropogenic activities and natural causes. Natural causes include fossil salt deposits, the weathering of salty parent materials, deposition of salts by water or wind, and the tidal flow of sea water or groundwater inflow in coastal lands. Anthropogenic activities that lead to degradation through salinization may include irrigation with saline water, poor drainage and irrigation management, replacement of perennial vegetation with annual crops (which changes water relationships), seepage of canal water, over-extraction of groundwater, over-use of agrochemicals, and using waste effluents in irrigation systems [50,51]. The type of soil salinity is indicated by measures including electrical conductivity (EC), soil pH, and soil sodium content. Sodium content is given as either sodium adsorption ratio (SAR), a measure of how much sodium is on the soil exchange sites relative to calcium and magnesium, and exchangeable sodium percent (ESP), a measure of how much of the total cation exchange sites are occupied by sodium. Salt affected soils have distinguishing features, such as the accumulation of salts on the soil surface, poor structure due to dispersion of clays, and others, as presented in Figure 5, Figure 6, Figure 7 and Figure 8.

Salt-affected soils can be classified geographically into coastal and inland salt-affected soils based on the Indian approach. Coastal saline soils are classified as saline soils and acid–saline soils based mainly on soil pH and EC, whereas inland salt-affected soils are classified into saline, sodic, and saline–sodic based on the values of soil pH, EC, and SAR or ESP [53]. The major areas that have salt-affected soils globally include Asia (mainly China, India, Bangladesh, Indonesia, Iran, Iraq, and Pakistan), Africa (mainly in the north of Africa including Egypt, Morocco, Algeria and Tunisia), North and Central of America (e.g., the western USA and Canada, and Mexico), South America (e.g., Argentina, Brazil, Chile, and Paraguay), Europe (mainly in Hungary, France, and Romania), and Australia [53]. The main features of saline/alkaline soils may include the growth of halophyte plants like purslane, the accumulation of salts on the soil surface, nutrient deficiency due to nutrient imbalances, plant dehydration, disease pressure due to decreased resistance, etc.

Salt-affected soils have several impacts on both soil and cultivated plants. Salinity stress is a complex process that negatively influences nearly all of a plant’s biochemical and physiological processes. As a result, crop productivity is decreased due to inhibition of plant growth, reduced biomass, and its yield, decline in shoots, leaves, flowers, and seeds, low water and nutrient uptake efficiency, induced-DNA damage, oxidative stress due to a high content of reactive oxygen species (ROS), inhibition of photosynthesis and cellular hydration, and accumulation of toxic ions, mainly Na^+^ [52,53,56,57,58,59]. Impacts on the soil itself include loss of structure, dispersion of organic matter, antagonism of nutrient update, increased soil erosion rate (due to high soil dispersibility and decrease shear stress), increased flooding rate (due to higher runoff because of low soil permeability), ecological imbalances (due to changes in vegetation including halophytes, bushes, mesophytes, and trees), and may cause problems for human health because of frequent malaria outbreak and other diseases [53,60,61]. Under salinized environments, many mechanisms could be adapted to make plants more tolerant to salinity, including (1) adaption through ionic homeostasis and osmotic adjustment (proline, betaine, etc.), (2) adaption through ROS scavenging (enzymatic and non-enzymatic antioxidants), (3) adaption through salt exclusion, removing and pumping salt out of root cells, and (4) adaption through salt secretion, leaf succulence, photosynthesis protection, and reduction of water loss in shoots [55].

Salt-affected soils can negatively affect crop productivity causing huge losses in both yield and its economic value. Thus, proper management strategies must be adopted to reduce stressful conditions on cultivated crops and to protect the soils from the devastating and deleterious impacts of this stress using combinations of the following approaches (Table 1):Application of Ca-sources like gypsum [53],Phytoremediation using halophytes [62,63],Application of biofertilizers [64],Nano-remediation using nanomaterials [65],Applying organic materials like biochar [66,67],Selecting proper crop genotypes [68,69],Using integrated fertilization [70],Maintaining soil water level by using proper fertilization/irrigation [71],Selecting efficient irrigation systems [72], andSoil management through techniques like tillage and mulching [73].

It is essential to utilize sustainable approaches to reduce the deleterious impacts of salinity stress, as reported by several published articles such as El Sabagh et al. [74]; Farid et al. [75]; Leal et al. [76]; Naz et al. [77]; Khan et al. [78] (Figure 9 and Figure 10).

**Table 1 plants-11-02392-t001:** Some published studies on managing salt-affected soils using different nanomaterials and biofertilizers.

Used Nanomaterials/Amendment	Cultivated Plant	Properties of Used Soil	Main Findings of This Study	Refs.
I. Applied nanomaterials				
Nano-Zn, nano-Si (30, 25 nm and 50, 2.5 mg L^−1^, resp.)	Rice (*Oryza sativa*, L.), var. Giza 178	Clayey, EC = 7.6 dS m^−1^, SAR = 14, ESP = 22.5%	Improved saline sodic soil by integrated management of both nano-Zn, and nano-Si in addition to using straw-filled ditches	[79]
Nano-ZnO at levels of 1 and 2 g·L^−1^ (40–50 nm)	Faba bean (*Vicia faba* L.), var. Sakha 1	Clayey, pH = 8.43, EC = 7.48 dS m^−1^, SAR = 16.2, ESP = 18.6	Application of nano-ZnO compost and S was integrated to reclaim saline–sodic soils	[80]
Green nano-silica (150 and 300 mg L^−1^)	Banana (*Musa* spp.)	Sandy irrigated with groundwater (EC = 4.12 dS m^−1^)	Green nano-silica improved the productivity and quality in sandy soil with saline irrigation	[81]
MgO-NP at 50 and 100 µg ml^−1^ as foliar application	Sweet potato (*Ipomoea batatas* L.) cv. Beauregard	Sandy loam, EC = 7.56 dS m^−1^, pH = 7.65, ESP = 10.66%	Co-applied effective micro-organisms and/or MgO-NPs improved plant tolerant to osmotic stress by increase osmolytes level, K^+^ content	[65]
Nanoparticles (Si-Zn-NPs) and plant growth-promoting microbes (PGPMs)	Soybean (*Glycine max* L.) cv. Giza 111	Clayey, pH = 8.23, EC = 5.52 dS m^−1^, ESP = 16%	PGPMs and nanoparticles (Si-Zn-NPs) promoted soybean productivity, and seed quality under water deficit stress	[82]
Foliar NPs-Si (12.5 mg L^−1^) and bio-Se-NPs (6.25 mg L^−1^)	Rice (*Oryza sativa* L.), Giza 177 and Giza 178	Clayey, pH = 8.20, EC = 7.20 dS m^−1^, SOM = 1.62%	Applied nano-nutrients (NPs-Si and NPs-Se) improved the yield components and mitigated harmful salinity stress	[83]
**II. Applied biofertilizers/organic fertilizers**				
Extracts of moringa leaves, licorice roots, ginger (2.0%)	Wheat (*Triticum aestivum* L.), cv. Misr 1	Clayey, pH = 8.13, EC = 13.20 dS m^−1^, ESP = 15.08%	Proline and enzymatic antioxidants (CAT, SOD) after treating with vermicompost and sprayed with moringa extract	[84]
PGPR, some strains of both *Rhizobium* and *Bacillus*	Faba Bean (*Vicia faba* L.), cv. 716	Clayey, pH = 8.24, EC = 5.52 dS m^−1^, SOM = 1.19%, ESP = 20%	Foliar PGPR and potassium silicate maintain soil quality and increased productivity of plants irrigated with saline water (3.5 dS m^−1^)	[85]
PGPR, namely some strains of *Azospirillum* and *Bacillus*	Wheat (*Triticum aestivum* L.), cv. Misr 1	Clay loam, pH = 8.58, EC = 9.09 dS m^−1^, SOM = 1.48%, ESP = 18%	Collaborative impact of PGPR and compost on soil properties, and physiological–biochemical attributes of wheat under water deficit stress	[86]
Bacterial inoculation (plant growth-promoting rhizobacteria)	Maize (*Zea mays* L.) cv. HSC 10	Clayey, pH = 8.22, EC = 7.33 dS m^−1^, ESP = 21.27%	Phosphor-gypsum and PGPR are effective approach for ameliorating the negative stress of salinity on maize plants	[87]
Foliar spray of folic acid (FA), ascorbic acid (AA), and salicylic acid (SA)	Potato (*Solanum tuberosum* L.) cv. Spunta	Loam, pH = 7.71, EC = 7.14 dS m^−1^, SOM = 0.79%	Foliar AA (200 mg L^−1^) was most effective in enhancing plant tolerance to salinity stress	[88]
Gypsum and mycorrhizal fungi inoculation (AMF)	Wheat (*T. aestivum* L.), cv. Sakha 94; maize (*Z. mays* L.), cv. Hybrid 368	Heavy clay, pH = 8.32, EC = 7.09 dS m^−1^, ESP = 19.35%, SOM = 1.16%	Combination of applied gypsum and AMF inoculation was an effective approach to ameliorate and alleviate the hazardous effects of soil salinity and sodicity on cultivated plants	[89]
PGPR (*Azospirillum brasilense* and *Bacillus circulans*); potassium silicate	Wheat (*T. aestivum* L.), cv. Misr 1, Gemmeza 12, and Sakha 95	Clayey texture, pH = 8.28, EC = 7.71 dS m^−1^, SOM = 1.75%	Combined application activated soil enzymes (i.e., urease and dehydrogenase); boosted soil microbial activity; enhanced plant growth at studied stress	[90]
Biochar (husks of rice and maize) and foliar applied potassium humate	Onion (*Allium cepa* L.), cv. Giza 20	Clay loam, pH = 8.35, EC = 11.14 dS m^−1^, SOM = 1.51%	Dual application of biochar and K-humate was sustainable, an effective, eco-friendly strategy under water stress	[91]

Abbreviations: soil electrical conductivity (EC); sodium adsorption ratio (SAR), soil organic matter (OM), (EC), and cation exchange capacity (CEC) was in cmol_c_ kg^−1^, respectively.

## 5. Sustainable Production of Nanoparticles

The sustainable production of nanoparticles (NPs) can be achieved using biological approaches (i.e., microorganisms and plants). This biosynthesis of nanoparticles is also called the green production of nanoparticles. The types of nanoparticles produced and the method of production may differ from plants to microbes (Table 2). Several publications have reported on the biosynthesis of nanoparticles and different environmental conditions that control this production (e.g., [92,93,94,95,96,97,98,99,100,101]). 

The role of nanomaterials in sustaining the soil and its fertility is a crucial issue that has been explored by many researchers, especially under soil pollution or degradation conditions (e.g., [38,62,102,103,104]). Microbial synthesis of NPs is considered a sustainable approach for nano-bioremediation of the environment, because these NPs can be non-toxic, clean, and eco-friendly, and this method allows renewable materials to be used for metal reduction and NP-stabilization [102]. 

## 6. Nano-Restoration of Soil Fertility

Greater crop productivity is essential to provide global food security with a growing global population. Without appropriate sustainable plant nutrition, this higher crop production will not be possible to achieve while providing ecological balance [105]. Nanotechnology is a promising approach to support and sustain agricultural production by tailoring nano-fertilizers, nano-pesticides, or nano-biofertilizers in an eco-friendly manner to meet the specific needs of cultivated crops [105]. Several applications of nanotechnology in agriculture are shown in Figure 11. The definition of nanotechnology, methods of producing nanoparticles (NPs) or nanomaterials (NMs), and benefits and risks of NPs in the environment can also be found in Figure 11. A risk–benefit analysis should be undertaken before applying any new nano-formulation to agricultural use [105]. 

Nanoparticles can be synthetized using three methods: physical, chemical, and biological (Figure 12). When microorganisms and plants are used in nanoparticle synthesis it is called green synthesis technology, which is considered cost-effective, biologically safe, and eco-friendly [95]. Microorganisms such as bacteria [111], algae [112], and fungi or mushrooms [34] have the ability to produce and bio-convert inorganic metal ions into nano-sized compounds. Some of these nano-particles can be used for sustainable agriculture [113]. These biological methods mainly depend on the synthesis of nanoparticles via both extra-cellular and intra-cellular enzymatic activities and intrinsic metabolic processes [95]. 

Restoration of soil fertility in soils degraded by problems such as pollution, salinization, desertification, etc. is a crucial global issue. Thus, sustainable approaches for soil restoration are needed for soil health and regenerative agriculture [114,115]. The sustainable remediation of degraded soils using nano-remediation or nano-restoration of soil fertility depends mainly on characterization of both soil and used nanomaterials [103,116,117]. The fate and behavior of nanomaterials in soil during these restoration processes depends on the applied nanomaterials, soil solution, and other properties of the soil being remediated [118]. 

## 7. Nanoparticle–Plant–Microbe Nexus for Restoring Soil Fertility 

Agroecosystems, which include soil, are complex, open, and dynamic systems. There are several interactions between different compartments of these systems (Figure 13). Components of an agroecosystem include soil, water, plants, microbes, animals, and humans. All these components continuously interact. Natural and/or anthropogenic NPs, like other agrochemicals or pollutants, are included in these interactions. Both positive and negative interactions, from a crop production perspective, are possible. Soil may behave as a natural sink that receives and stores all agrochemicals, NPs, pollutants, nutrients, etc. Therefore, NPs may be taken up by plants, accumulate in the soil system as nano-pollution, or be used in the process of remediation of pollutants from soil and water through nano-remediation [118,119,120]. The following sub-sections present some different interactions among these agroecosystem compartments with a focus on nanoparticles, cultivated plants, and soil microbes. How these interactions can be adopted to restore soil fertility is the main question posed in these sub-sections. 

### 7.1. Soil–Plant–Water Interactions

The soil–plant–water system includes the impacts of soil physical, chemical, and biological properties, cultivated plants (depends on plant species), and soil water (i.e., soil solution), in which soil reactions occur. The main factors that control the soil–plant–water nexus are linked to the different properties of these components. This system is important for controlling soil health and fertility (Figure 14). For example, if soils are polluted, different nutrient and pollutants pathways will be present. The management of polluted soil depends on the kind of pollutant(s) and its (their) concentration, plant species, and applied soil amendments such as biochar (e.g., [64,66,67,71]), fly ash [121], and organic amendments [66,84]. 

### 7.2. Soil–Plant–Microbe Interactions

The rhizosphere is an important area that is characterized by several interactions between plant root exudates (i.e., amino acids, organic acids, and carbohydrates) and soil microbes. There is a continuous transformation of soil organic matter to available nutrients [52]. These nutrients are available to both plants and microbes, which increases soil fertility. Without this interaction between plants and their associated soil microbes, many nutrients would not be available for plants (Figure 15). Several agrochemicals have the potential to be harmful to the soil–plant–microbe system and must therefore be managed expeditiously [47]. Alternatives to chemical fertilizers should be explored and developed, such as plant growth promoters (plant growth-promoting bacteria or PGPB and plant growth-promoting rhizobacteria or PGPR) and biofertilizers, to create eco-friendly and sustainable agricultural systems [47]. The soil–plant–microbe nexus behaves differently under different environmental conditions like salinity [52], alkalinity [84], pollution [124], mining soils [39], and integrated mineral regulation under the plant–microbe system [125]. The role of plants and microbes in the restoration of soil health and fertility can be attributed mainly to soil nutrient accumulation and vegetation restoration through root exudates as well as boosted nutritional metabolism of plants via microbial enzymes [39]. A strong relationship between soil fertility and the interaction between plants/microbes and soil has been reported in the literature, especially using different indicators of soil health and/or some soil fertility indices such as biochemical index of soil fertility (Kucharski et al., 2009) [126]. This index could be calculated from the soil enzyme activities (e.g., nitrogenase, urease, dehydrogenases, alkaline and acid phosphatase) and depends on cultivated plant species, soil characterization, and the time of soil sampling (Symanowicz et al., 2021 [127]).

### 7.3. Soil–Water–Nanoparticle Interactions

Nanoparticles can move in the soil–water–atmosphere system through several pathways (Figure 16). The main interactions of nanoparticles in soil may include NPs–plant–microbe interactions, NPs–plant–soil–water–microbe interactions, and NPs–plant–microbe–pollutant interactions (Figure 17). In the rhizosphere there are strong interactions between plant root exudates and NPs, which can alter plant root exudates to facilitate transformations of nanoparticles [119]. The main interactions of NPs in soil may include the following pathways: mobility of NPs in soil, aggregation, and disaggregation of NPs in soil, NPs’ dissolution or chelation in soil, toxic impacts of NPs on soil organisms, the chemical speciation and release of metals in soil, and the sorption of NPs on soil particles [118,120]. Factors affecting fate and transport of NPs in soil are particle characterization (size, surface, content, etc.), solution conditions (pH, ionic strength, OM, pollutants), soil properties (physical, chemical, and biological characteristics), and flow characterization (flow rate, flow conditions, including constant head and constant flow) [119,120]. 

Factors affecting NP mobility in soil may include NP-properties (mainly bare or coated NPs, composition, and its concentration), soil properties (e.g., pH, ionic strength, SOM, soil texture, soil moisture content), effects on NP–bioavailability, NP–reactivity and fate, and the soil microbial community and activities [118,120]. Chemical speciation and release of NP–metals may depend on soil properties (e.g., soil pH, SOM, other elements concertation, etc.), sorption of NPs on soil particles (shape of NPs, their mobility, aggregation in soil, etc.), and NPs–toxicity on plants and soil microbes [118,120]. 

### 7.4. Soil–Plant–Water–Microbe Interactions

Dynamic and intensive interactions among plants, soil, and microbes occur in the rhizosphere, which supports plant productivity via regulating availability of nutrients, controlling plant diseases, and signaling secondary metabolites (Figure 18). These previous mechanisms could be used by plants to deal with environmental stresses. Plants can shape the structure of the microbes in the rhizosphere with their root exudates, and many soil microbes (e.g., bacteria, fungi, actinomycetes, etc.) thrive in the rhizospheric niche [129,130,131]. 

Several mechanisms of these interactions and the processes driving changes in soil microbes are still unknown, especially in the case of polluted soil and interactions with nanomaterials or nanoparticles. Research to date has focused on the role of soil microbes as relates to plant mineral nutrition [130], improving the efficiency of fertilizers [131], remediation of degraded mine soils [131], soil microbial communities affecting plant productivity under climate change [132], plants and microbes for restoring soil fertility [133], and plants and microbes for restoration of natural vegetation [134]. Restoring soil fertility utilizing the soil–plant–microbes nexus depends on the biological activity of plants, root exudates, and soil microbial activities and their enzymes, as well as the presence of pollutants or agrochemicals. 

### 7.5. Soil–Plant–Microbe–Nanoparticle Interactions

Study of above-ground and below-ground microbes, which may link to plant physiological functions and immunity, is critical in the restoration of soil fertility [133]. Nano-restoration is a soil remediation process using nanomaterials or nanoparticles. The use of nanotechnology to restore polluted or degraded soils has received attention in many published articles such as Rajput et al. [124]. This approach may include alleviating plant stress using nanomaterials [135], using halophytic nanoparticles in remediating saline soils [62], or application of nanoremediation for environmental cleanup [103]. The NPs prevalent in water, soil, and the atmosphere may interact with plants, leading to accumulation in those plants. The NPs then enter the food chain, which can cause problems for human health [119]. Possible interactions among the soil–plant–microbe–NPs nexus are shown in Figure 19. 

The possible ways in which engineered NPs could accumulate in soil include NPs released during their synthesis from the atmosphere to soil, NPs released during their use in soil or wastewater treatment, and NPs disposal in landfills or by incineration [100]. The soil microbiome, and other soil organic matter forms, like humic substances, play a vital role in plant nutrition and yield. This role may change when NPs, pollutants or agrochemicals are introduced to soil. In the soil system, a complex, dynamic and open system allows many interactions among its components, including plants, microbes, agrochemicals, pollutants, and NPs. Therefore, studies are needed to cover the many open questions concerning these interactions. 

## 8. General Discussion

This review focuses mainly on nano-restoration of degraded, including polluted, soils. The role of soil microbes and nano-materials in the rhizosphere in promoting (or restricting) the productivity of cultivated plants under different growth conditions has also been discussed. The following section will address the meaning of soil fertility, the meaning of soil degradation, how soil restoration can be achieved in the era of nanotechnology, and the potential negative side of nano-restoration. 

Soil fertility refers to the ability of a soil to support plant productivity. This includes the supply of nutrients, but also includes other aspects of soil chemical, physical, and biological properties [136]. The over-application of agrochemicals (e.g., mineral fertilizers, pesticides including insecticides, fungicides, and herbicides) can upset the delicate equilibrium in the soil system, negatively affecting fertility. Therefore, agricultural management is vital to reduce and reverse soil degradation. It is important to sustain fertile soils, including well balanced nutrients content, adequate soil organic matter, and diversity and abundant soil life [136].

Soil degradation is a global issue, that has resulted from several aspects of soil management. Nano-restoration represents a new approach to reverse these degradation processes, but it needs more study to fully understand the positive and negative aspects of nano-restoration. This work investigated nano-restoration and its role in sustaining soil fertility relying heavily on a pictorial presentation. Nanomaterials such as nano-silica, carbon-based nanomaterials, polymer-based nanomaterials, and metal-based nanoparticles can be used in soil nano-remediation [103]. These nanomaterials can be released into soil unintentionally or intentionally, during their application and use, during their preparation, or when they are disposed of in the soil such as in landfills [100]. 

The ability of soil nanoparticles to be toxic to soil microbes, plants, and then humans through the food chain are a concern. The published literature documents negative impacts of nanomaterials applied to soil such as zero-valent iron (nZVI), which can negatively affect soil microbial activity when over applied [137]. On the other hand, nZVI is effective at removing nitrate from groundwater [138], which is a positive use. Combined application of CeO_2_-NPs and bacteria is an effective approach in alleviating Cr-toxicity in sunflower plants [139]. Cadmium stress may be alleviated through application of nano-Se and microbes including *Alphaproteobacteria*, *Anaerolineae, Bacteroidia*, *Deltaproteobacteria*, *Gammaproteobacteria*, and *Gemmatimonadetes* in the rhizosphere [140]. The combined nano- and bio-remediation approach has also been applied to Pb-diesel fuel co-polluted soil using nZVI [141]. The nano-toxicity threat to soil micro-organisms, cultivated plants, and then human health has been confirmed from the over-use of nanofertilizers (e.g., ZnO- and CuO-NPs) (e.g., [107,142,143]). The suggested mechanism of nano-toxicity on cultivated plants is oxidative stress and damage in biochemical, morpho-physiological, and molecular insights in edible plants, which may cause serious impacts on human health [144]. The impacts nanomaterials on agricultural soil microbiota also were reported [145]. On the other hand, the suggested mechanisms of applied nanomaterials that may improve plant salinity tolerance could be presented in Figure 20. These mechanisms depend on the kind of nanomaterials, the applied dose, and plant species [100,146,147,148]. 

Based on the available information covered in this review, there are several open questions that should be answered. These include, but not limited to: -What are the roles of microorganisms in their interplay with plants and NPs for restoring soil fertility? -What are the evolutionary and ecological basis of microbe–plant–soil interactions? -What are the dynamics of microbe–plant interactions and their link to plant growth and soil conditions under the umbrella of nano-restoration? -What are the broader impacts of the interactions of microbe–plant–soil agroecosystems or agricultural productivity under soil degradation? -To what extent will the dynamics of microbe–plant interactions differ in the case of polluted or otherwise degraded soils? -What are the crucial roles of soil microbes for plant mineral nutrition and soil fertility in the presence of pollutants?

## 9. Conclusions

Given the increase in global population, an increase in global food production and other necessities supplied by soil is an urgent issue. About 25% of global soils are degraded to a point that has created a severely reduced ability to meet human needs. Thus, this review discusses degraded soils and their restoration. Nano-restoration is the main topic of this pictorial review, which presents the use of nanomaterials for nano-remediation of polluted soils with a focus on the role of soil microbes and cultivated plants through the nanoparticle–plant–microbe nexus and its interactions. However, the over application of nanomaterials during the nano-remediation process or during agricultural management may lead to nano-toxicity to cultivated or edible plants that moves through the food chain, culminating in negative health impacts for humans. Therefore, the nano-restoration of degraded soils should follow regulations designed to avoid health problems for humans and the environment. The possible interactions between nanoparticles, plants, and microbes in degraded or polluted soils need additional study for a better understanding of nano-restoration to sustain soil fertility. All possible interactions among all soil components besides water, cultivated or grown plants, any agrochemicals in soil, including pollutants, fertilizers, or nanoparticles. The expected fate and behavior of agrochemicals in soil–water–plant–microbe system still needs more and more investigations under different conditions especially under climate change. Day by day, the role of soil microbes has increased, and several negatives of these microbes have been changed into a positive impact, particularly in the field of pharmaceuticals. The integrative role of soil microbes, plant, and nanoparticles has gained great concern recently from researchers and companies all over the world. The interactive systems among soil microbes, plant, pollutant, or agrochemical and nanomaterials still have a very rich area of scientific research for sustaining the soil fertility.

## Figures and Tables

**Figure 1 plants-11-02392-f001:**
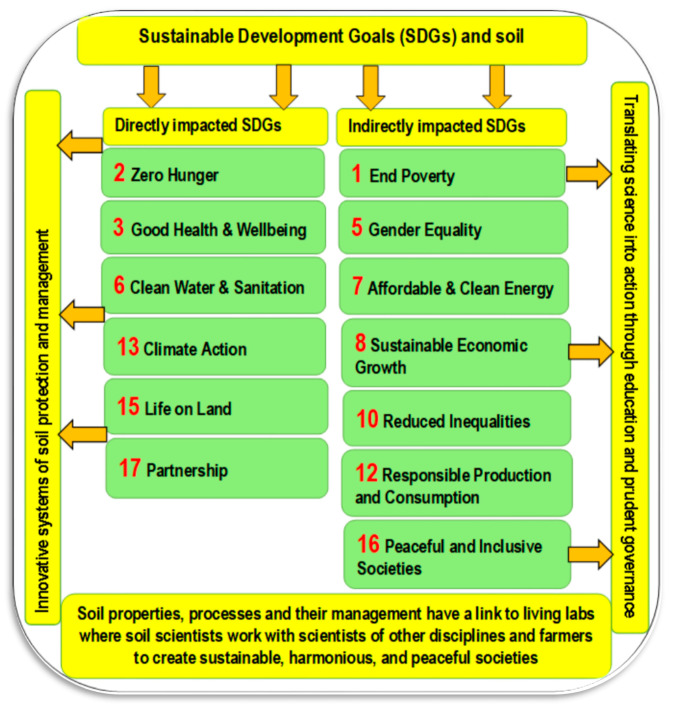
Sustainable Development Goals and their relationships with soil management (Sources: Lal et al. [32]).

**Figure 2 plants-11-02392-f002:**
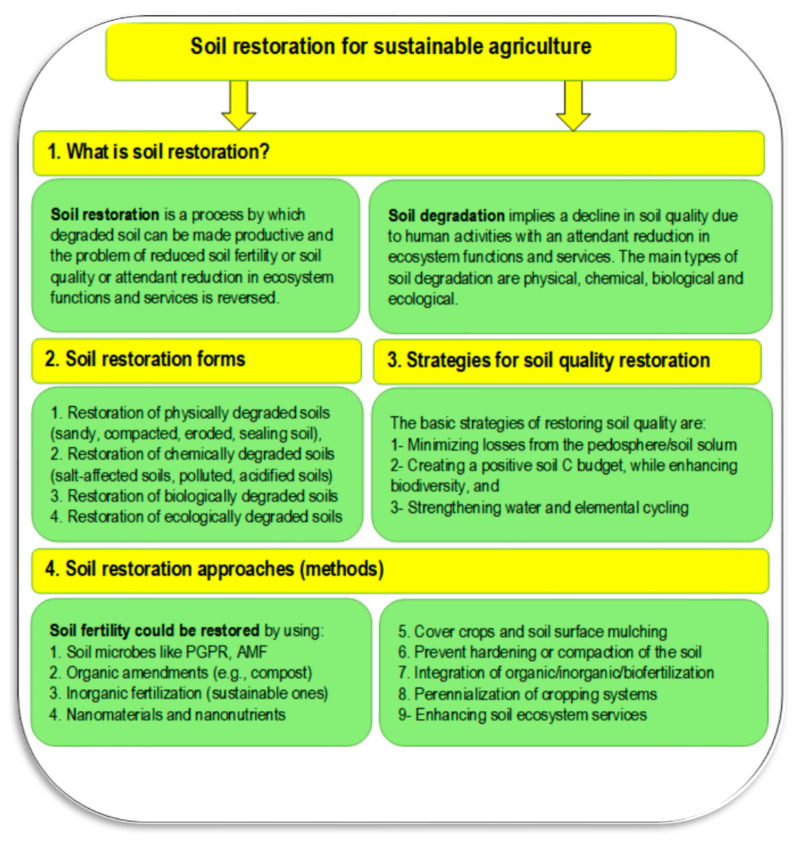
Soil restoration for sustainable agriculture including the forms, the approaches, and strategies (Sources: [31,38,39]). Abbreviations: Plant growth promoting rhizobacteria (PGPR), arbuscular mycorrhizal fungi (AMF).

**Figure 3 plants-11-02392-f003:**
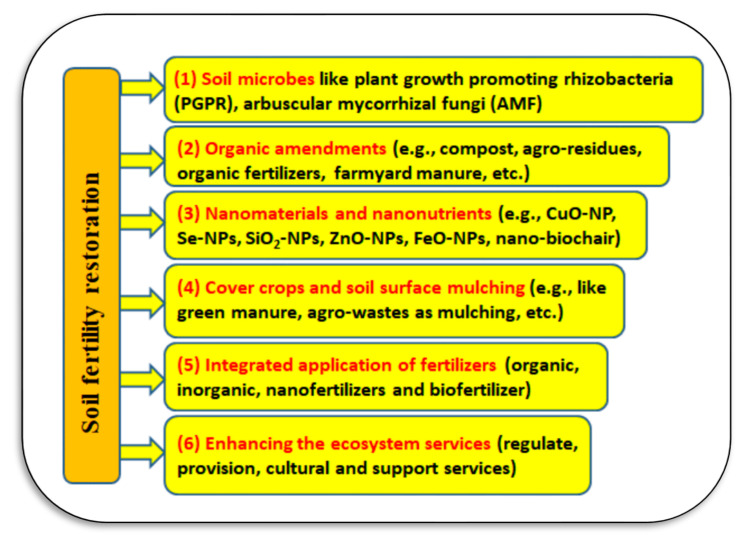
Different applying approaches for soil fertility. Sources: [38,39].

**Figure 4 plants-11-02392-f004:**
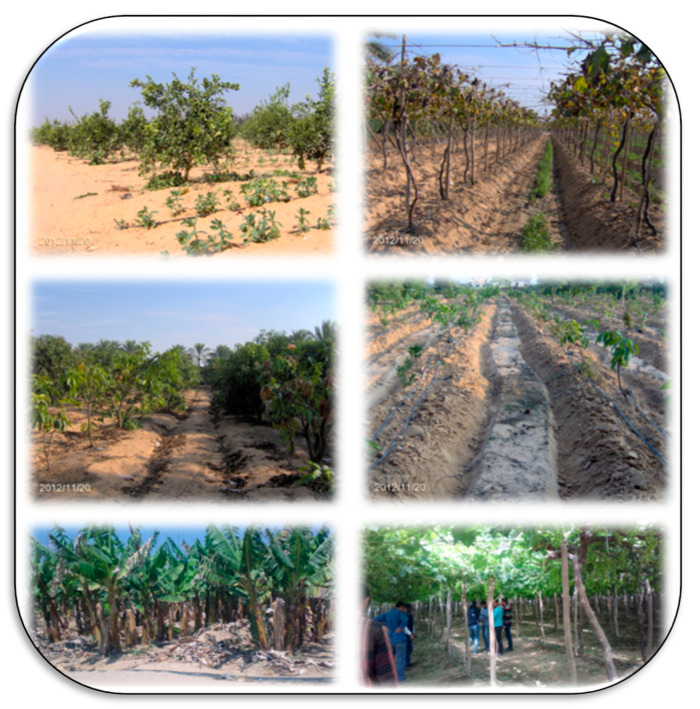
Cultivation of sandy soils is a great challenge facing the arid and semi-arid regions because of low fertility and low ability to hold water. These photos represent cultivation of sandy soil with different horticultural crops in Egypt, including citrus, grapes (**higher** photo left from saline sandy and right), mango (**middle** photos, which represent saline sandy soils), and banana (**lower** photo left). Photos by El-Ramady.

**Figure 5 plants-11-02392-f005:**
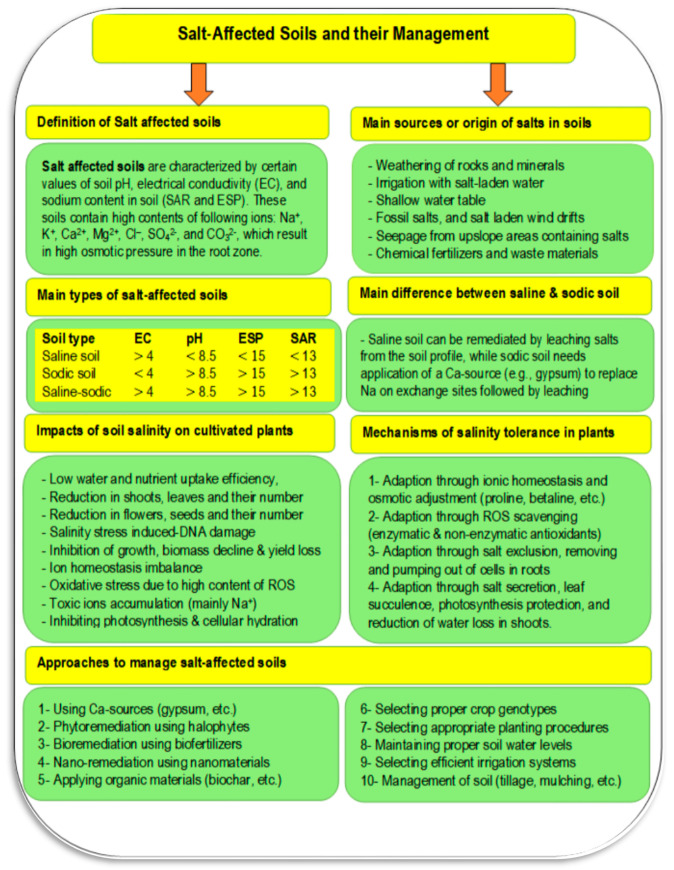
Definition of salt-affected soil, sources of salts, different impacts of soil salinity on cultivated plants, mechanism of salinity tolerance, and different management approaches (Sources: [52,53,54,55]).

**Figure 6 plants-11-02392-f006:**
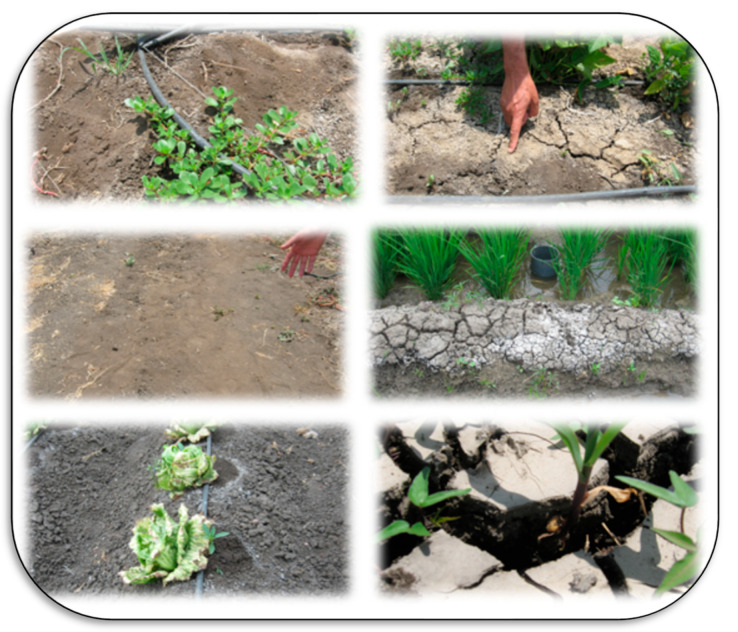
Some common features of saline-alkaline soils at the experimental farm of Kafrelsheikh Uni. (Egypt), which represent in sabkha on the soil surface and growing the purslane plants (the first higher 2 photos beside the middle photo **left**), the accumulation of salts during rice growing in saline soil in the middle photo **right**, and the lower photos (**left**) general view to saline soil during cultivating lettuce under drip irrigation and deep cracks due to heavy clay content (the lower **right** photo). Photos by El-Ramady.

**Figure 7 plants-11-02392-f007:**
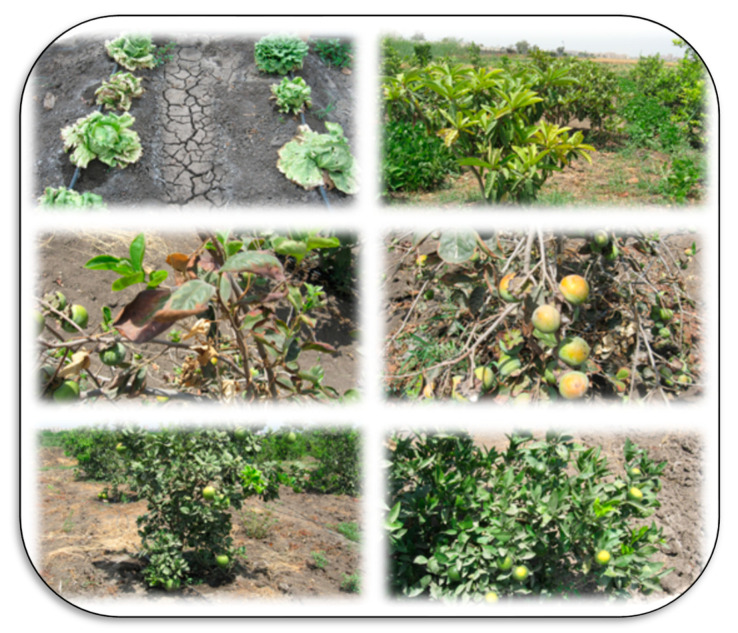
Production of horticultural crops under arid climatic zones and salinity stress in salt-affected soils at the experimental farm of Kafrelsheikh University (Egypt). Many crops physiological and nutritional problems (mainly nutrient imbalances, dehydration, disease pressure due to decreased resistance) can be seen on the cultivated crops from top to bottom; lettuce, sugar-apple tree (**top** photos), persimmon (the **middle** photos), and citrus (**lower** photos). Photos by El-Ramady.

**Figure 8 plants-11-02392-f008:**
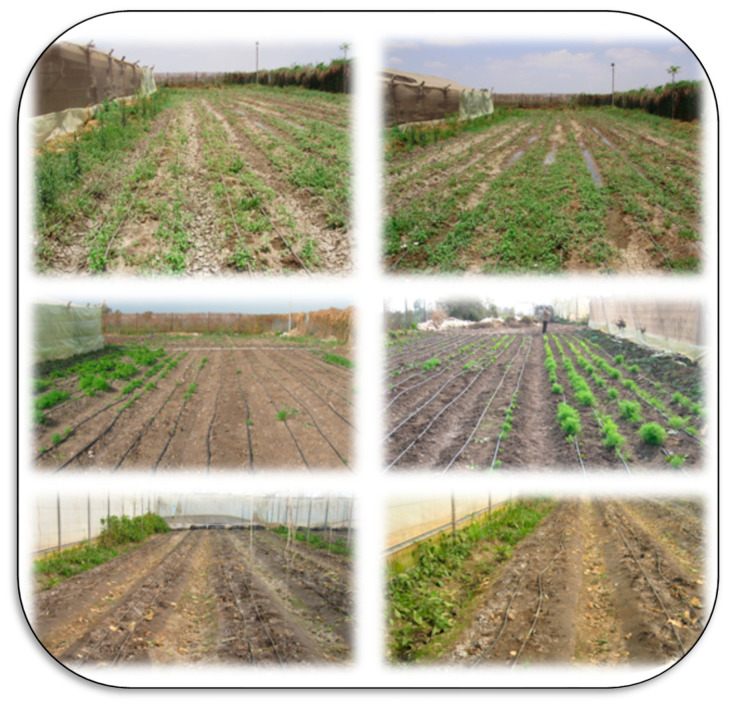
Salt-affected soils have general characteristics, including the accumulation of salts on the surface of the soil, missing plants due to high soil salinity in the field or under greenhouse conditions, high water table content due to poor drainage, especially in traditional greenhouses, and high temperature, which increases evaporation from the soil surface and thus accumulation of salts on the soil surface. Photos by El-Ramady.

**Figure 9 plants-11-02392-f009:**
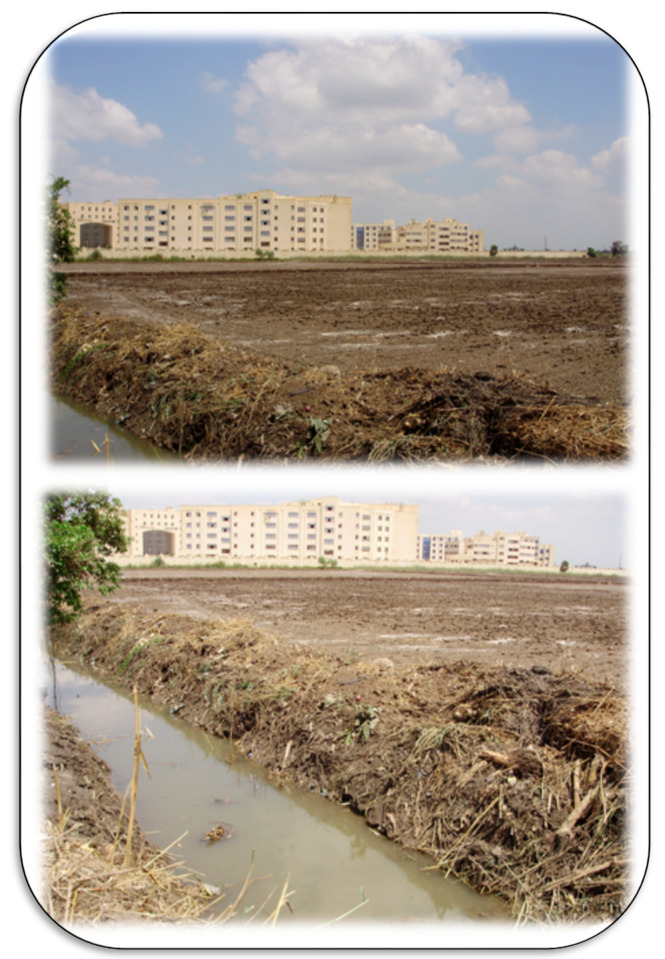
Saline–sodic soils in Kafrelsheikh, Egypt, could be managed using the application of gypsum (seen as the white spots on the soils in the photos). Cleaning the agricultural canals and/or drains is common at the experimental farm of Kafrelsheikh University to avoid harmful impact of Na in such soils, which is necessary to provide good drainage and reduce anthropogenically-induced salinization of the soils. Photos by El-Ramady.

**Figure 10 plants-11-02392-f010:**
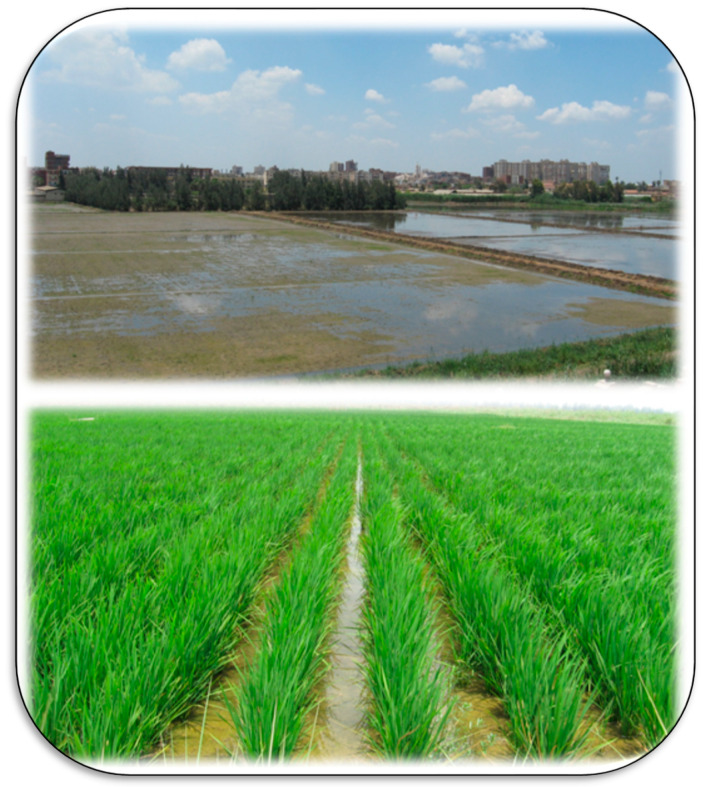
Cultivation of paddy rice is very important in managing salt-affected soils in the Kafr El-Sheikh region (Egypt), which depends on the flooding irrigation to overcome soil salinity in this area. Photos by El-Ramady.

**Figure 11 plants-11-02392-f011:**
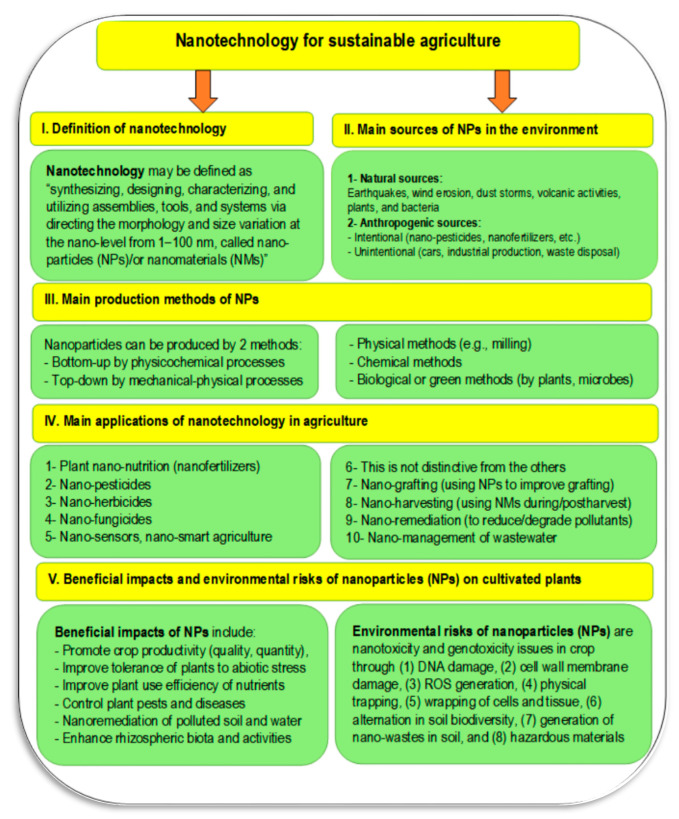
The relationship between nanotechnology and sustainable agriculture including different applications, methods of nano-production, and their benefits and challenges in agricultural application (Sources: [106,107,108,109,110]).

**Figure 12 plants-11-02392-f012:**
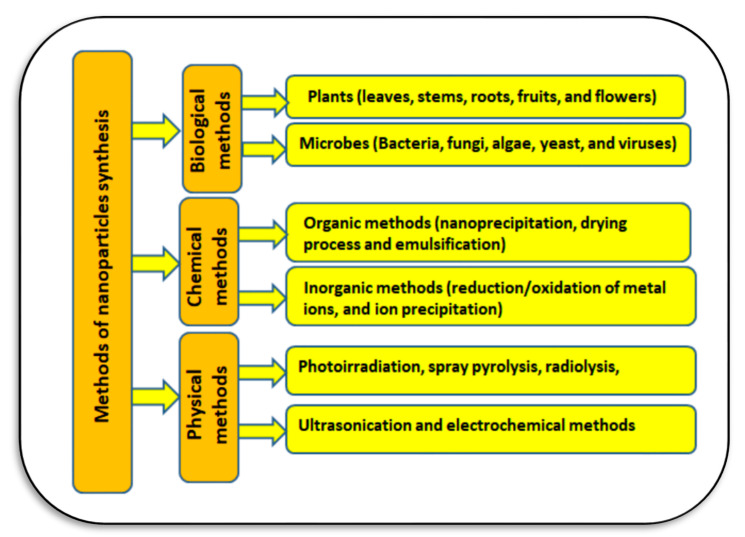
The common methods used in nanoparticle synthesis and their classification into physical, chemical, and biological methods (Sources: [62,95,112]).

**Figure 13 plants-11-02392-f013:**
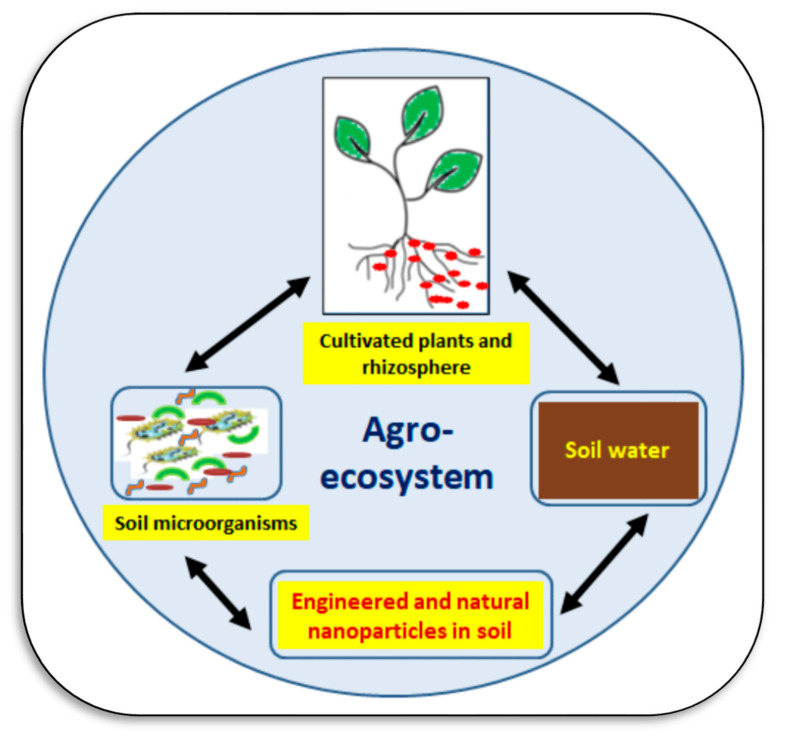
A simplified general overview of the interactions among soil microbes, cultivated plants, soil water (soil solution), and different kinds of nanoparticles (natural and atherogenic) in an agroecosystem. All these components interact together in soil, with many positive and negative impacts on soil fertility. Sources: [118,119,120].

**Figure 14 plants-11-02392-f014:**
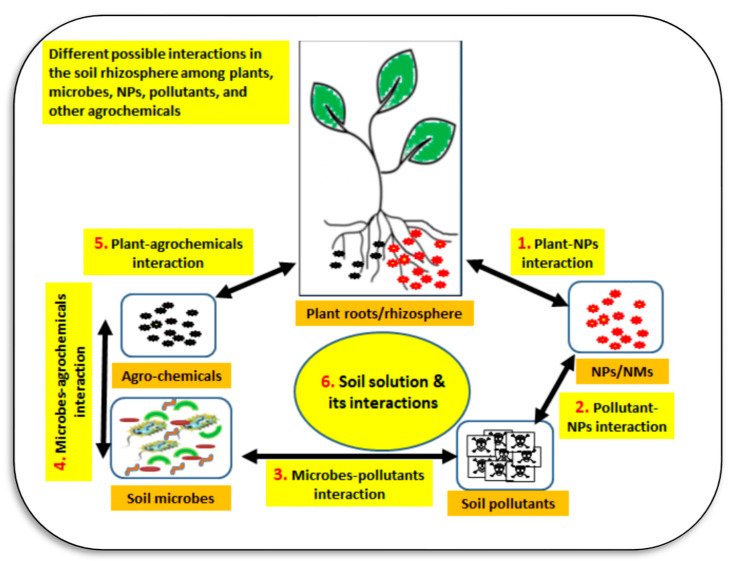
The soil solution is the medium where several interactions among soil microbes, plants, pollutants, agrochemicals, and nanoparticles occur in agroecosystems. These interactions can happen in the soil, with many positive and negative impacts on soil health and fertility. Sources: [118,122,123,124].

**Figure 15 plants-11-02392-f015:**
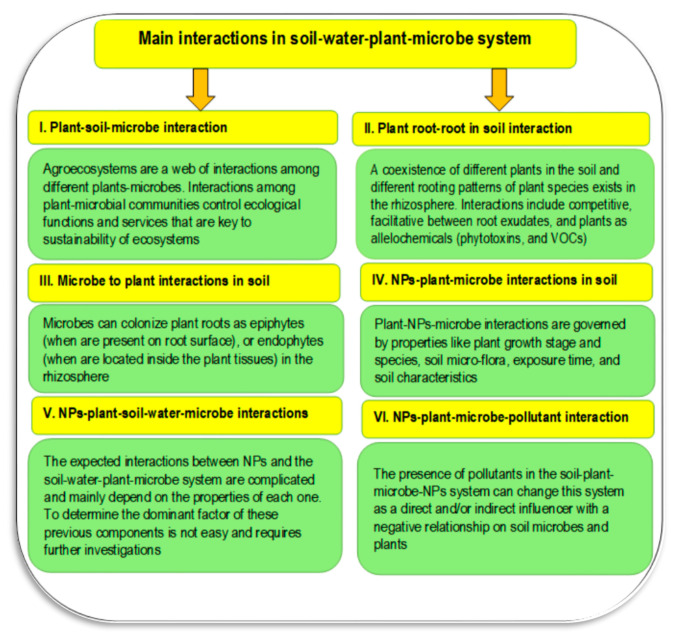
The possible interactions that can happen in the soil–water–plant–microbe system. **Sources**: [39,46,119]). **Abbreviation**: Volatile organic compounds (VOCs).

**Figure 16 plants-11-02392-f016:**
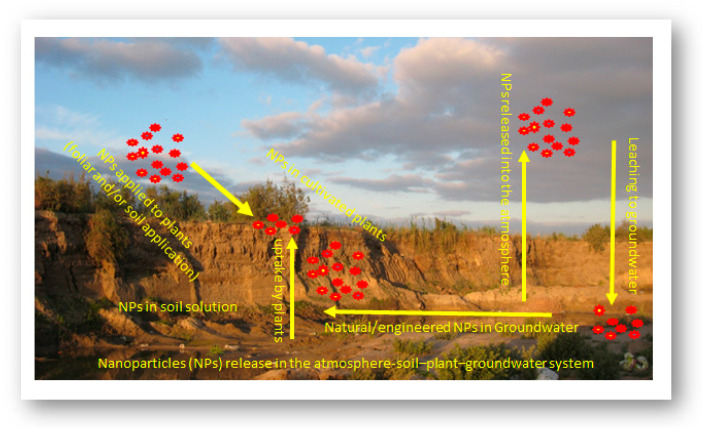
The fate and behavior of nanoparticles (NPs) released into the atmosphere–soil–plant–groundwater system (adapted from [128]).

**Figure 17 plants-11-02392-f017:**
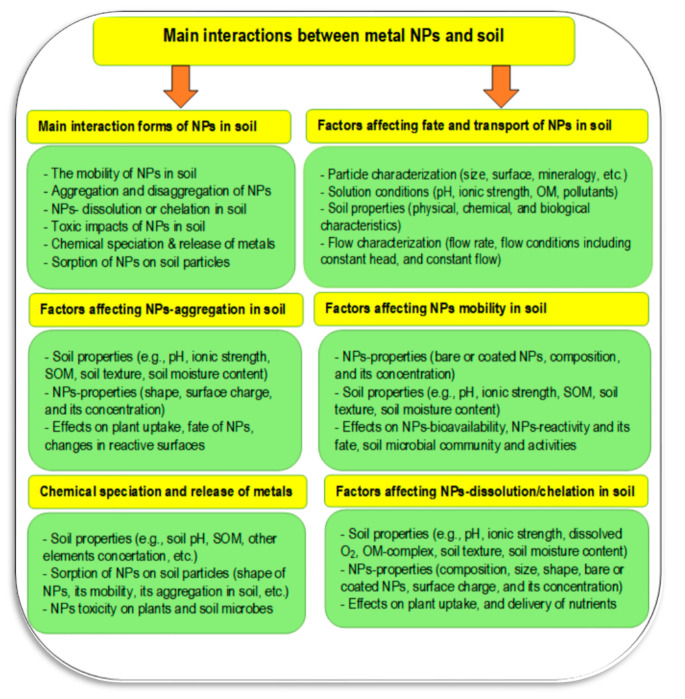
The main fate and interactions of nanoparticles in soil including various components of the soil system (Sources: [118,120]).

**Figure 18 plants-11-02392-f018:**
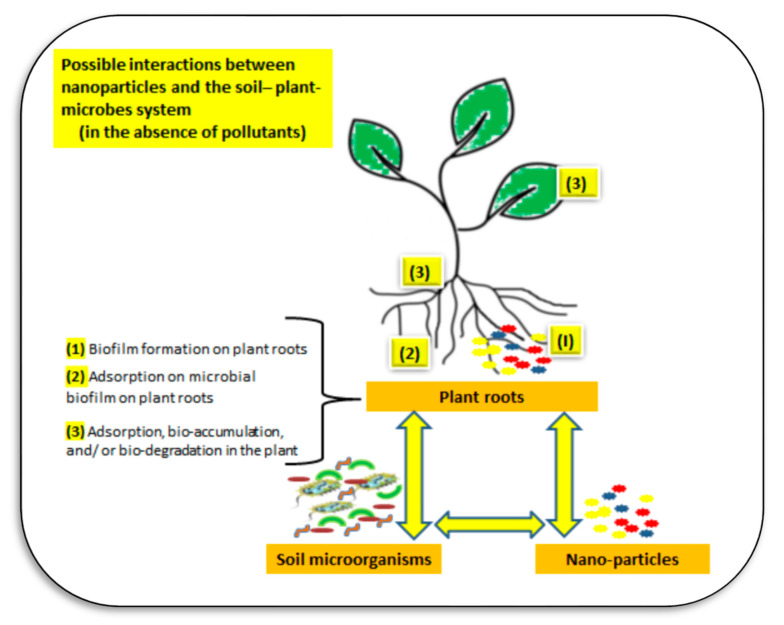
The interactions between soil microbes, plants, and nanoparticles involve pathways in the soil like forming biofilms on plant roots, adsorption, and bio-degradation in plants, but many mechanisms are still unknown and need more investigation. Sources: [118,122,123,124].

**Figure 19 plants-11-02392-f019:**
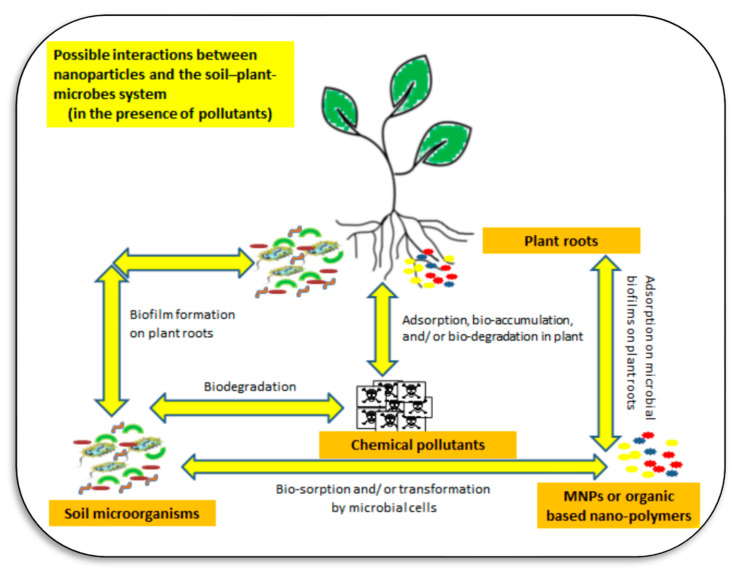
Different soil rhizospheric interactions between plant roots, agrochemicals (mainly mineral fertilizers and pesticides), organic/inorganic pollutants, soil micro-organisms (bacteria, fungi, actinomycetes, viruses), nanoparticles (NPs)/and or nanomaterials (NMs), and their interactions in the soil solution. Relationships between plants, microbes, and nano-materials during restoration of contaminated soil are still open questions that need more investigation. Sources: [118,122,123,124].

**Figure 20 plants-11-02392-f020:**
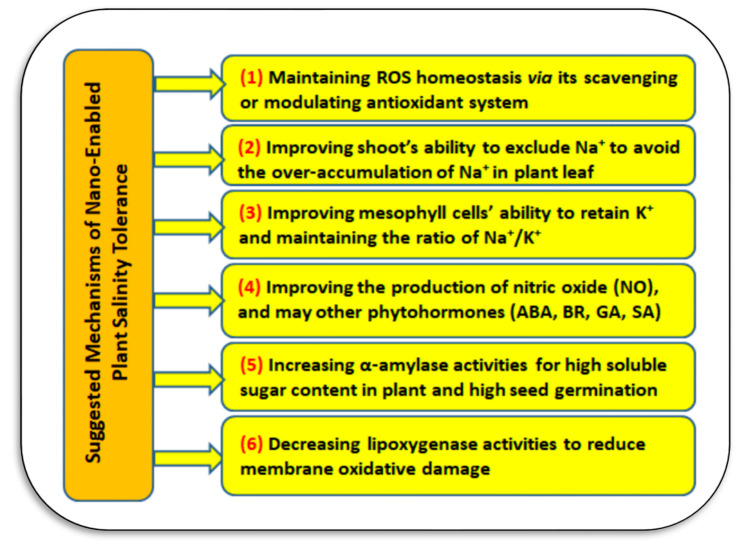
The suggested mechanisms of applied nanomaterials that may improve plant salinity tolerance depending on the kind of nanomaterials, the applied concentration, and plant species (**Sources:** [100,146,147,148]). **Abbreviations:** Reactive oxygen species (ROS), Abscisic acid (ABA), brassinosteroids (BR), gibberellin (GA), and salicylic acid (SA).

**Table 2 plants-11-02392-t002:** A comparison between microbes and plants in producing nanoparticles.

Item (s) of Comparison	Microorganisms	Plants
Method of synthesis	The biological/green method or biosynthesis	The biological or green methods
Which plant tissue or microbe can use?	Bacteria, fungi, yeast, viruses, cyanobacteria, and actinomycetes	Plant tissues (leaf, flower, seed, stem, root, peel, fruit) and plant extracts
location of production	Extracellular and intracellular	Extracellular and intracellular
Main mechanism	Extracellular biosynthesis by trapping metal ions on the cell wall and reducing them through secreted enzymes as reducing agents (e.g., acetyl xylan esterase)	Extracellular production of nano-particles using plant extracts (e.g., leaf, fruit, etc.) as capping agents in the production of nanoparticles, fast degradation of metal ions
	Intracellular biosynthesis by reducing metal ions into cell cytoplasm through metabolic reactions with enzymes (e.g., nitrate reductase), phytochemicals	Proteins, amino acids, vitamins, polysaccharides, polyphenols, terpenoids, organic acid
Factors affecting biosynthesis of nanoparticles	Medium pH, reaction time, temperature, and reactant content	Plant part (e.g., leaves, flowers, seeds, barks, fruits, and roots), plant species, extract content, temperature, metal in the salt, pH, and contact time
Main applications	Anti-cancer materials, cosmetics and medical appliances, antimicrobial, antipathogen, plant-growth stimulation, antifungal activity	Nano-sensors detect biomolecules, environmental factors, gene delivery cell labelling, magnetically responsive drug delivery, photothermal therapy

**Sources**: [92,95,96,99,101,102].

## Data Availability

Not applicable.

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
