# Peer review of "Nano-Restoration for Sustaining Soil Fertility: A Pictorial and Diagrammatic Review Article"

_plants, 2022, doi:10.3390/plants11182392_

Round 1

Reviewer 1 Report

The manuscript is suitable for publication in the journal.

Author Response

Reviewer 1#

Comments and Suggestions for Authors

The manuscript is suitable for publication in the journal.

Response: Many thanks!

Submission Date

23 August 2022

Date of this review

01 Sep 2022 18:02:32

Reviewer 2 Report

Comments and Suggestions for Authors

Nano-Restoration for Sustaining Soil Fertility: A Pictorial and Diagrammatic Review Article

The subject matter of the manuscript is loosely related to the Journal profile. However, in my opinion, you cannot write about plants without taking into account the soil. Therefore, I consider it appropriate to include this manuscript in Plants The presented manuscript deals with the current global problem. The review article presents a photographic and schematic review of the literature on nano-reconstruction as a soil fertility support. The effect of the nanoparticle-plant-microbe (NPM) compound in restoring soil fertility was described.

General remarks

A review of the most recent literature is an advantage of this manuscript. The wording of the manuscript is very clear.

An important source of organic matter is missing from this article. It is a waste brown coal with a low energy. (e.g. Effects of brown coal, sludge, their mixtures and mineral fertilisation on copper and zinc contents in soil and italian ryegrass (Lolium multiflorum Lam.). Fresenius Environmental Bulletin  2012, 21(4), 802-807. It can certainly be included in Table 1.

Subsection 7.2 – The https://doi.org/10.3390/agronomy11071335 has some useful contents to be added to the manuscript.

Specific comments

ü  Figure 5 – missing.

ü  Figures 3, 14, 15, 19 and 20 – source must be completed.

ü  Jacoby, R; Peukert, M.; Succurro, A.; Koprivova, A.; Kopriva, S. The Role of Soil Microorganisms in Plant Mineral Nutrition—Current Knowledge and Future Directions. Front. Plant Sci. 2017, 8, 1617. doi: 10.3389/fpls.2017.01617 – not cited in the manuscript.

The manuscript must be corrected according to the editorial requirements of the publisher Best regards,

Author Response

Reviewer 2#

Comments and Suggestions for Authors

Nano-Restoration for Sustaining Soil Fertility: A Pictorial and Diagrammatic Review Article

The subject matter of the manuscript is loosely related to the Journal profile. However, in my opinion, you cannot write about plants without taking into account the soil. Therefore, I consider it appropriate to include this manuscript in Plants The presented manuscript deals with the current global problem. The review article presents a photographic and schematic review of the literature on nano-reconstruction as a soil fertility support. The effect of the nanoparticle-plant-microbe (NPM) compound in restoring soil fertility was described.

Response: Many thanks for so kind words!

General remarks

A review of the most recent literature is an advantage of this manuscript. The wording of the manuscript is very clear.

Response: Many thanks, again!

An important source of organic matter is missing from this article. It is a waste brown coal with a low energy. (e.g., Effects of brown coal, sludge, their mixtures and mineral fertilisation on copper and zinc contents in soil and italian ryegrass (Lolium multiflorum Lam.). Fresenius Environmental Bulletin 2012, 21(4), 802-807. It can certainly be included in Table 1.

Response: Many thanks for your suggestion, but this table included the role of nanomaterials or biofertilizers in managing the salt-affected soils, which their properties were inserted in the 3rd column (mainly soil pH, salinity or EC, and SAR or ESP), as the title of Table 1 said:

Table 1: Some published studies on managing salt-affected soils using different nanomaterials and biofertilizers.

So, I can not use this article in this Table, because the Table is talking only on salt-affected soil, whereas the soil in the suggested paper is not salt-affected soil as below: This is the part of soil properties according you your suggested article:

  1. MATERIALS AND METHODS

A three-year pot experiment was conducted by a random method against a control (loamy sand - LS) in three replications. The soil material used in the experiment had the following physicochemical properties: pH in KCl mol dm-3 – 7.0, Corg. – 9.2 g kg-1, Nog. – 0.74 g kg-1, Hh – 29.2 mmol(+) kg-1, S – 15 mmol (+) kg-1, available P, K and Mg in soil (mg kg-1) – 59.0; 106.3 and 29.1, respectively.”

Subsection 7.2 – The https://doi.org/10.3390/agronomy11071335 has some useful contents to be added to the manuscript.

Response: Many thanks for your suggestion paper!

The suggested article was cited and some information was added to the revised MS as follows:

A strong relationship between soil fertility and the interaction between plants/microbes and soil has been reported in the literature, especially using different indicators of soil health and/or some soil fertility indices such as biochemical index of soil fertility (Kucharski et al. 2009). This index could be calculated from the soil enzyme activities (e.g., nitrogenase, urease, dehydrogenases, alkaline and acid phosphatase) and depends on cultivated plant species, soil characterization, and the time of soil sampling (Symanowicz et al. 2021).

Specific comments

Figure 5 – missing.

Response: Many thanks, it was a wrong in the text and corrected in the revised MS!

Figures 3, 14, 15, 19 and 20 – source must be completed.

Response: Many thanks for your suggestion!

The sources were added for each figure, thanks again as follows:

[118, 122-124]

Jacoby, R; Peukert, M.; Succurro, A.; Koprivova, A.; Kopriva, S. The Role of Soil Microorganisms in Plant Mineral Nutrition—Current Knowledge and Future Directions. Front. Plant Sci. 2017, 8, 1617. doi: 10.3389/fpls.2017.01617 – not cited in the manuscript.

Response: Many thanks, this ref. has become no. 129 in the revised MS, thanks again!

The manuscript must be corrected according to the editorial requirements of the publisher

Response: Many thanks, Sure, we are waiting any further comment from the editor, we are ready for correction of the revised MS to be ready for publication, thanks again!

Best regards,

Submission Date

23 August 2022

Date of this review

03 Sep 2022 19:41:06

Reviewer 3 Report

Dear authors, the manuscript Nano-Restoration for Sustaining Soil Fertility: A Pictorial and Diagrammatic Review Article presents an interesting concept. 

There are some changes I consider will increase the quality of your work.

General comments: consider to remove any personal expressions like "we". This will make your text look more professional. Do not repeat too often the same word, like "nexus", in consecutive sentences.

The overall length and the construction of the manuscript sections is good and presents well the topic. 

The conclusion section should be rewritten, in order to point a conclusion for each subsections of the manuscript. 

I like the idea and was a pleasure to read the manuscript.

Author Response

Reviewer 3#

Comments and Suggestions for Authors

Dear authors, the manuscript Nano-Restoration for Sustaining Soil Fertility: A Pictorial and Diagrammatic Review Article presents an interesting concept.

Response: Many thanks for so kind words!

There are some changes I consider will increase the quality of your work.

General comments: consider to remove any personal expressions like "we". This will make your text look more professional.

Response: Many thanks for comment, we removed all “we” from the MS!

Do not repeat too often the same word, like "nexus", in consecutive sentences.

Response: Many thanks for comment, as we can, because sometimes the meaning needs this word, and we can NOT remove it!

The overall length and the construction of the manuscript sections is good and presents well the topic.

Response: Many thanks for so kind words, again!

The conclusion section should be rewritten, in order to point a conclusion for each sub-sections of the manuscript.

Response: Many thanks for comment, the subsections of the MS were presented in the conclusion section as follows:

“The possible interactions between nanoparticles, plants, and microbes in degraded or polluted soils need additional study for a better understanding of nano-restoration to sustain soil fertility. All possible interactions among all soil components besides water, cultivated or grown plants, any agrochemicals in soil including pollutants, fertilizers or nanoparticles. The expected fate and behavior of agrochemicals in soil-water-plant-microbe system still needs more and more investigations under different conditions especially under climate change. Day by day, the role soil microbes has increased and several negatives of these microbes have been changed into a positive impact particularly in the field of pharmaceuticals. The integrative role of soil microbes, plant, and nanoparticles has gained a great concern recently from the researchers and companies all over the world. The interactive systems among soil microbes, plant, pollutant or agrochemical and nanomaterials still have a very rich area of scientific research for sustaining the soil fertility.

I like the idea and was a pleasure to read the manuscript.

Response: Many thanks for so kind words, thanks a lot!

Submission Date

23 August 2022

Date of this review

06 Sep 2022 12:34:37
